# Bisphenol A Exposure Changes the Transcriptomic and Proteomic Dynamics of Human Retinoblastoma Y79 Cells

**DOI:** 10.3390/genes12020264

**Published:** 2021-02-11

**Authors:** Chul-Hong Kim, Mi Jin Kim, Jinhong Park, Jinho Kim, Ji-Young Kim, Mi-Jin An, Geun-Seup Shin, Hyun-Min Lee, Jung-Woong Kim

**Affiliations:** Department of Life Science, Chung-Ang University, Seoul 06974, Korea; 01031e70067@gmail.com (C.-H.K.); mjkim831025@gmail.com (M.J.K.); shoopjh@cau.ac.kr (J.P.); jinhokim07@cau.ac.kr (J.K.); jykim@cau.ac.kr (J.-Y.K.); dksalwls333@gmail.com (M.-J.A.); rjstjq89@naver.com (G.-S.S.); lhmscb@naver.com (H.-M.L.)

**Keywords:** RNA-seq, alternative splicing, proteome, bisphenol A, apoptosis, retinoblastoma

## Abstract

Bisphenol A (BPA) is a xenoestrogen chemical commonly used to manufacture polycarbonate plastics and epoxy resin and might affect various human organs. However, the cellular effects of BPA on the eyes have not been widely investigated. This study aimed to investigate the cellular cytotoxicity by BPA exposure on human retinoblastoma cells. BPA did not show cytotoxic effects, such as apoptosis, alterations to cell viability and cell cycle regulation. Comparative analysis of the transcriptome and proteome profiles were investigated after long-term exposure of Y79 cells to low doses of BPA. Transcriptome analysis using RNA-seq revealed that mRNA expression of the post-transcriptional regulation-associated gene sets was significantly upregulated in the BPA-treated group. Cell cycle regulation-associated gene sets were significantly downregulated by exposure to BPA. Interestingly, RNA-seq analysis at the transcript level indicated that alternative splicing events, particularly retained introns, were noticeably altered by low-dose BPA treatment. Additionally, proteome profiling using MALDI-TOF-MS identified a total of nine differentially expressed proteins. These results suggest that alternative splicing events and altered gene/protein expression patterns are critical phenomena affected by long-term low-dose BPA exposure. This represents a novel marker for the detection of various diseases associated with environmental pollutants such as BPA.

## 1. Introduction

Bisphenol A (BPA) is widely used in the manufacturing of plastics, epoxy resin liners of canned foods, beverage containers, dental materials, and medical devices. BPA is easily eluted from these products and directly exposed to the human body via foods and drinks. Consequently, BPA has been detected in human blood, amniotic fluid, breast milk, and adipose tissue, resulting in alterations to fetal development, with associated increased risk of adverse health outcomes [1,2,3,4]. BPA is a xenoestrogen chemical with estrogen-like activity associated with adverse effects on the human endocrine systems. It is among the most debated examples of endocrine-disrupting chemicals (EDCs) [5]. For example, it has been reported that BPA exposure can reduce the number of mature oocytes and increase fertilization failure in females as well as decrease sperm production and quality in males [6,7,8]. Numerous studies have demonstrated that BPA is probably associated with type 2 diabetes, cardiovascular disorders, neurobehavioral diseases, and various cancer types [9]. In particular, several studies have investigated the adverse effects of BPA on the central nervous system owing to its lipophilic properties, which can lead to its accumulation in the brain [10,11]. Moreover, intrauterine BPA exposure can lead to sex-specific epigenetic alterations in the brain, even at low doses [12]. Based on this accumulating evidence, many countries have banned BPA use. However, BPA is still detected in a variety of environments, including water, dust, sewage, and air [13]. Therefore, there are increasing public concerns about its harmful effect, as humans are constantly exposed to such environmental pollutes. Most related studies have focused on the hazardous effects of BPA on the reproductive system and development secondary to its anti-estrogen activity. However, little is known about the effects of BPA on the eyes.

The eyes are exposed directly to the outside environment; therefore, they are designed to be protected from foreign substances, such as dust, wind, water, and bright light. However, the eyes are vulnerable to external atmospheric and environmental toxins. According to the World Health Organization, air pollution consists of various particulates, including BPA, mercury, and particulate matter (PM) [14]. There is increasing evidence that air pollution can directly affect the eyes. For example, chronic ambient exposure to NO_2_ and PM 2.5 disrupt ocular surface homeostasis through decreasing goblet cell density and mucin 5AC (MUC5AC) mRNA expression [15]. Additionally, exposure to fine PM pollution decreases the abundance of corneal epithelial cells and conjunctival goblet cells in C57BL/6 mice; long-term exposure also causes morphological changes to the mouse ocular surface [16]. Exposure to air or water contaminated by toxic pollutants can damage the eyes, resulting in ocular injury and impaired vision. BPA is also among the components of polluted air. Baba et al. demonstrated that BPA causes abnormal eye development and morphology, such as exerted lens development, by disrupting Notch signaling in Xenopus laevis. Although studies have demonstrated associations between air pollution and eye injury, it remains unknown whether BPA affects the human visual system, especially retinal neurons.

Based on the possible harmful effects of BPA on the human eye, we decided to identify the effects of BPA on gene and protein expression profiles using human Y79 retinoblastoma cells. We examined the effect of BPA on the cell cycle and apoptosis, as well as on the transcriptome profile using RNA sequencing, and the proteome profile using proteomic analysis with MALDI-TOF-MS.

## 2. Materials and Methods

### 2.1. Chemicals and Reagents

All the cell culture reagents used in the present study were produced from Welgene (Seoul, Republic of Korea). Bisphenol-A (BPA) (cat. no. 47889) was purchased from Sigma-Aldrich (Oakville, ON, Canada). BPA 1M stock solution was prepared and diluted in 100% DMSO. Cells were treated with various concentration of BPA (20, 40, 80, 100, 500, and 1000 μM) in DMEM containing 10% FBS. MTS assay kit (cat. no. G3581) was obtained from Promega Corporation (Madison, WI, USA). Antibodies against Ki-67 (cat. no. ab15580) was purchased from Abcam, cyclin B1 (cat. no. ab32053) was obtained from Santa Cruz Biotechnology (Dallas, TX, USA), and cyclin D1 (cat. no. 2978S) were obtained from Cell Signaling Technology Inc. (Beverly, MA, USA). Goat anti-rabbit or mouse secondary antibody (cat. no. 111-095-144) was obtained from Jackson ImmunoResearch (West Grove, PA, USA).

### 2.2. Cell Culture

Human retinoblastoma Y79 cells were purchased from the ATCC (Manassas, VA, USA). Cells were maintained in RPMI 1640 supplemented with 10% fetal bovine serum (FBS), 1% penicillin/streptomycin at 37 °C in a humidified, 5% CO_2_ atmosphere. Cells were seeded at a density of 4 × 10^6^ cells per well in 100 mm cell culture dish or 6-well culture plate. After 24 h incubation, cells were treated with different concentration of BPA in media supplemented with 10% FBS without antibiotics for 24 or 48 h. The cells in 100 mm cell culture dishes or 6-well culture plates were gently washed with phosphate-buffered saline and were trypsinized. Cells were processed according to the different experimental analysis.

### 2.3. Cell Viability Assay

For the determination of cell viability, Y79 cells were seeded in 48-well plates at the density of 4 × 10^4^ cells per well and incubated for 24 h with growth medium. The cells were treated with the different concentrations of BPA (20, 40, 80, 100, 500, and 1000 μM) in 100 μL culture medium containing 10% FBS. The solvent 0.1% DMSO treated cells were served as control. After treatment of indicated time periods, cell viability was performed using CellTiter 96^®®^ AQueous One Solution Cell Proliferation Assay (Promega, cat. no. G3581). Twenty microliters of MTS (3-(4, 5-dimethylthiazol-2-yl)-5-(3-carboxymethoxyphenyl)-2-(4-sulfophenyl)-2*H*-tetrazolium) solution was added and the cells were incubated for 1 h at 37 °C. The outcome cell viability was monitored via a Multiscan Go microplate reader (Waltham, MA, USA) by measuring absorbance at 490 nm.

### 2.4. Morphological Change and Live/Dead Cell Assay

To assess the effect of BPA on cytotoxicity, we were used the cells seeded in 6-well culture plate. Cells were treated with the indicated BPA in a different time period. Cells were stained with a mixture of two dye: 0.3 μM calcein-AM, 3 μM of EthD-1 (Life Technologies, Carlsbad, CA, USA). Dyes were prepared immediately in 1× PBS before use and added to the cells. After 30 min incubation in dark condition, dye solution was not washed out, and images were captured using Nikon Eclipse TE300 inverted fluorescence microscope (Nikon Corp., Tokyo, Japan).

### 2.5. Annexin V-FITC/Propidium Iodide (PI) Apoptosis Assay

To assess the apoptosis, the cells were harvested and centrifuged at 3000 rpm for 3 min. Cell proportions were detected using FITC Annexin V Apoptosis Detection Kit I (BD pharmingen, La Jolla, CA, USA) according to the manufacturer’s instructions. The cell pellets were washed with 1× PBS and resuspended in 100 μL of Annexin V binding buffer. Cells were labeled with 5 μL of Annexin V-Alexa Fluor 488 and/or 5 μL of PI. After 15 min incubation in a dark, 400 μL of Annexin V binding buffer was added to wash the Annexin/PI stained cells. A minimum of 10,000 cells per sample was analyzed using BD Accuri™ C6 Plus (BD FACS, San Jose, CA, USA).

### 2.6. Cell Cycle Analysis

After BPA treatment, cells were fixed for 1 h in 70% ethanol at 4 °C and centrifuged at 3000 rpm for 3 min, and the pellets were washed with ice-cold PBS. Cell pellets were then resuspended in 0.1 mL of PBS containing 100 μg/mL RNase A for 1 h at 37 °C. Then, 50 μg/mL of propidium iodide (Sigma-Aldrich) was added and then incubated for 15 min at RT in a dark. PI-stained cells were analyzed by BD Accuri™ C6 Plus (BD FACS, San Jose, CA, USA). At least 10,000 cells were used for each analysis, and the results were displayed as histograms. The percentage of cell distribution in Sub-G_1_, G_0_/G_1_, S, and G_2_/M phase were measured, and the results were analyzed by the BD Accuri™ C6 Plus software for cell cycle profile.

### 2.7. Immuno-Staining and Flowcytometry Analysis

Cells were fixed for 5 h in 1% paraformaldehyde at 4 °C and centrifuged at 3000 rpm for 3 min. Cell pellets were washed once with ice-cold 1× PBS. Cells were incubated with the designated primary antibodies for 30 min in a dark. Then, cells were washed and incubated with the secondary FITC-labeled antibody for 20 min at RT in a dark. Expression of the specific target protein was analyzed by BD Accuri™ C6 Plus (BD FACS, San Jose, CA, USA). At least 10,000 cells were used for each analysis, and the results were displayed as histograms. The percentage of the cell population was measured by FITC positive cells.

### 2.8. Total RNA Isolation and RNA-Sequencing

Cells were seeded in 100 mm cell culture dishes at the density of 2 × 10^6^ cells per well and incubated for 24 h with growth medium. Total RNA was extracted using TRIzol. The purity was assessed by Agilent 2100 Bioanalyzer using the RNA 6000 Nano Chip (Agilent Technologies, Santa Clara, CA, USA). RNA quantitation was performed using Thermo scientific Multiskan GO microplate reader with μDrop™ Plate (Waltham, MA, USA). The construction of RNA-sequencing library was performed using Truseq RNA Sample preparation kit v2 (Illumina, Inc., San Diego, CA, USA, cat. no. RS-122-2002). In brief, 100 ng of total RNA from each sample was incubated with poly-T oligo-attached magnetic beads to isolate poly-A tailed mRNA followed by mRNA fragmentation. The cleaved RNA fragments were constructed a double-stranded cDNA. Then, the double-stranded library was purified by using AMPure XP beads to remove all reaction components. The end repair, a base addition, adapter ligation, and PCR amplification steps were performed according to the manufacturer’s instructions. Libraries quality and quantity were checked by an Agilent 2100 Bioanalyzer using the High Sensitivity DNA Chip (Agilent Technologies, Santa Clara, CA, USA). Then, the cDNA libraries were used for paired-end 75 sequencing using an Illumina NextSeq 500 system (Illumina, Inc., San Diego, CA, USA).

### 2.9. Bioinformatical Analysis

The entire analysis pipeline of RNA-seq was coded by using R (ver. 3.6), which was controlled by systemPipeR (ver. 1.18.2). Raw sequence reads were trimmed for adaptor sequence and masked for low-quality sequences using systemPipeR. Transcript quantification of RNA-seq reads was performed with GenomicAlignments (ver.1.20.1) by reads aligned to Ensemble v95 Homo Sapiens transcriptome annotation (GRCh.38.95) using Rsubread (ver. 1.24.6). The FPKM (Fragments Per Kilobase of transcript per Million mapped reads) values were calculated using “fpkm” function from DESeq2 (ver. 1.24.0) that processed on the robust median ratio method and transcript reads were normalized by “voom” function from Limma (ver. 3.40.6). To analyze a transcript as differentially expressed, EdgeR (ver. 3.26.7) calculates the results based on the normalized counts from entire sequence alignments. Significantly differentially expressed transcripts having greater than fold change in raw FPKM value > 2 and *p*-value < 0.01 cases in all experimental comparison were selected and used for further analysis. Gene annotation was added by the online database using Ensemble biomaRt (ver. 2.40.4), and visualization was performed by using R base code and gplots (ver. 3.0.1.1).

### 2.10. Protein Sample Preparation and 2-DE

The cultured animal cell pellets were washed twice with ice-cold PBS and sonicated for 10 s by Sonoplus (Bandelin electronic, Berlin, Germany), which was homogenized directly by motor-driven homogenizer (PowerGen125, Fisher Scientific, Pittsburgh, PA, USA) in sample lysis solution composed with 7M urea, 2M Thiourea containing 4% (*w*/*v*) 3-[(3-cholamidopropy) dimethyammonio]-1-propanesulfonate (CHAPS), 1% (*w*/*v*) dithiothreitol (DTT) and 2% (*v*/*v*) pharmalyte and 1 mM benzamidine. Proteins were extracted for one hour at room temperature with vortexing. After centrifugation at 15,000× *g* for one hour at 15 °C, insoluble material was discarded, and soluble fraction was used for two-dimensional gel electrophoresis. Protein concentration was assayed by Bradford method. IPG (immobilized pH gradient) dry strips (4–10 NL IPG, 24 cm, Genomine, Korea) were equilibrated for 12–16 h with 7 M urea, 2 M thiourea containing 2% 3-[(3-cholamidopropy) dimethyammonio]-1-propanesulfonate (CHAPS), 1% dithiothreitol (DTT), 1% pharmalyte and, respectively, loaded with 200 μg of sample. Isoelectric focusing (IEF) was performed at 20 °C using a Multiphor II electrophoresis unit and EPS 3500 XL power supply (Amersham Biosciences, Pittsburgh, PA, USA) following manufacturer’s instruction. For IEF, the voltage was linearly increased from 150 to 3500 V for 3 h for sample entry followed by constant 3500 V, with focusing complete after 96 kVh. Prior to the second dimension, strips were incubated for 10 min in equilibration buffer (50 mM Tris-Cl, pH 6.8 containing 6 M urea, 2% SDS, and 30% glycerol), first with 1% DTT and second with 2.5% iodoacetamide. Equilibrated strips were inserted onto SDS-PAGE gels (20 × 24 cm, 10–16%). SDS-PAGE was performed using Hoefer DALT 2D system (Amersham Biosciences, Pittsburgh, PA, USA) following manufacturer’s instruction. Two-dimensional gels were run at 20 °C for 1700 Vh. Then, 2D gels were silver stained as described by Oakley et al. (Anal. Biochem. 1980, 105:361–363), but fixing and sensitization step with glutaraldehyde was omitted. Quantitative analysis of digitized images was carried out using the PDQuest (version 7.0, BioRad, Hercules, CA, USA) software according to the protocols provided by the manufacturer. The quantity of each spot was normalized by total valid spot intensity. Protein spots were selected for the significant expression variation deviated over two-fold in its expression level compared with control or normal sample.

### 2.11. MALDI-TOF-MS and Protein Quantification

For protein identification by peptide mass fingerprinting, protein spots were excised, digested with trypsin (Promega, Madison, WI, USA), mixed with alphpa cyano-4-hydroxycinnamic acid in 50% acetonitrile/0.1% TFA, and subjected to MALDI-TOF analysis (Microflex LRF 20, Bruker Daltonics) as described Fernandez J et al. (Electrophoresis 19: 1036–1045). Spectra were collected from 300 shots per spectrum over *m*/*z* range 600–3000 and calibrated by two point internal calibration using Trypsin auto-digestion peaks (*m*/*z* 842.5099, 2211.1046). Peak list was generated using Flex Analysis 3.0. Threshold used for peak-picking was as follows: 500 for minimum resolution of monoisotopic mass, 5 for S/N. The search program MASCOT, developed by The Matrixscience (http://www.matrixscience.com/ (accessed on 5 February 2021)), was used for protein identification by peptide mass fingerprinting. The following parameters were used for the database search: trypsin as the cleaving enzyme, a maximum of one missed cleavage, iodoacetamide (Cys) as a complete modification, oxidation (Met) as a partial modification, monoisotopic masses, and a mass tolerance of ± 0.1 Da. PMF (Peptide mass fingerprinting) acceptance criteria is probability scoring.

### 2.12. Western Blot

The collected cells were lysed with a total lysis buffer containing 1% Triton X-100, 150 mM NaCl, 50 mM Tris-HCl, pH 7.5, 0.1% sodium dodecyl sulfate (SDS), 1% NP-40, and protease inhibitor (1 mM PMSF, 1X PIC) by vortexing. The cell suspensions were incubated on the ice for 3 min and centrifuged at 13,000× *g* at 4 °C for 10 min. The protein samples were quantified by Bradford protein assay kit (Bio-Rad) and electrophoresed on a 10% SDS-PAGE and transferred to a nitrocellulose membrane (Protran™; Whatman, Maidstone, UK). The membrane was blocked with 5% skim milk in a solution of TBS-T (137 mM NaCl, 20 mM Tris–HCl (pH 7.6) and 0.1% Tween-20) and incubated with the primary antibodies at 4 °C for overnight. The protein samples were analyzed by Western blotting using the suitable antibodies to detect protein expression. Mouse monoclonal antibody against HYOU1 (1:1000; sc-398224) and mouse monoclonal antibody for β-actin (1:1000; sc-47778) were purchased from Santa Cruz Biotechnology (Santa Cruz, CA, USA).

### 2.13. Statistical ANALYSES

Values are presented as mean ± SEM of experiments performed in triplicate. Data were analyzed by two-way ANOVA followed by Tukey’s multiple comparison test using GraphPad Prism software version 5.01 (San Diego, CA, USA). Differences between groups were considered to be significant at *p* < 0.05.

## 3. Results

### 3.1. The Effect of BPA on Cell Growth and Death in Human Retinoblastoma, Y79

To investigate the cellular physiological effects of low-dose BPA exposure on retinal cells, Y79 cells were treated with various BPA concentrations for 24 and 48 h. The viability of the Y79 cells was tested via the MTS assay. Cells were treated with BPA (20–1000 μM) for 24 and 48 h (Appendix A). BPA treatment significantly reduced the viability of Y79 cells at high concentrations (500 and 1000 μM). Compared with the untreated group, treatment with BPA for 24 h decreased the viability of Y79 cells, with an IC_50_ value of 550 μM that further decreased to 323 μM at 48 h (Appendix A). The viability test was further examined using the live/dead assay with Y79 cells. Concordance with the MTS assay results, BPA treatment significantly increased the number of dead cells at with 500 μM exposure, but no effect was observed with 40 μM exposure (Figure 1a), suggesting that below concentrations of 100 μM, BPA has no cytotoxic effects on human retinoblastoma Y79 cells.

Next, we determined whether BPA affected cell proliferation and cell cycle progression. For this, cells were treated with a lower range of concentrations of BPA (20–100 μM) then subjected to the MTS assay for 24 and 48 h (Appendix A). As shown in Figure 1b, at 48 h, treatment with 40 µM BPA did not induce the expression of Ki-67, which is a cell proliferation marker or cell cycle progression (Figure 1b). Ki-67 expression increased after BPA treatment for 24 h in a dose-dependent manner, but this effect ceased after longer exposure (Appendix A). Cell cycle progression was also not altered by BPA exposure (Appendix A). Consistent with the cell cycle results, BPA did not affect the expression of the cell cycle markers, cyclin B1 and cyclin D1 (Appendix A). To further investigate whether BPA causes apoptosis in Y79 cells, cells were treated with 40 μM of BPA. After incubation, cells were stained with Annexin-V and propidium iodide (PI) to detect apoptosis using flow cytometry analysis. BPA treatment did not affect the number of apoptotic cells (Figure 1c). We also assessed the effect of BPA on caspase-3 activity. The active forms of caspase-3 were not altered by BPA treatment (Figure 1c). Compared with no treatment, BPA treatment had no effect on apoptosis (Figure 1c, Appendix A). These results indicated that BPA did not have cytotoxic effects on Y79 human retinoblastoma cells.

### 3.2. Dynamic Transcriptome Profiling of BPA-Treated Y79 Retinoblastoma Cell Line

To examine the effects of BPA exposure on transcriptional changes in Y79 cells, we selected a concentration (40 μM) and incubation time (48 h) of BPA that was associated with minimal effects on cell death and physiology, and the cells were subjected to directional RNA-seq analysis (datasets publicly available at GSE 146255). For this purpose, total RNA was extracted, and libraries for mRNA-sequencing were constructed from two independent samples exposed to BPA or a control. After filtering the RNA-seq data (FPKM > 1, at least in one group), a comprehensive list of 40,035 transcripts was obtained (Appendix A). The two datasets indicated concordant transcriptome dynamics in BPA-exposed and control groups. To test congruency among biological replicates, the principal component analysis (PCA) using DESeq revealed that BPA treatment accounted for the largest variance among all datasets generated using the RNA-seq platform. The cluster between the biological replicates of each experimental group was shown together, confirming high reproducibility between each replicate (Figure 2a). To identify the expression pattern of mRNAs during BPA treatment in Y79 cells, heat maps were constructed to profile the entire transcriptome differences at 48 h. Unsupervised hierarchical clustering analysis based on Pearson’s correlation of averaged and log2 of normalized FPKM values of the BPA-treated and control groups showed a decisive shift in the BPA-exposure transcriptome in the form of upregulated and downregulated transcripts (Figure 2b). Application of DESeq with a conservative approach to the RNA-seq data obtained from the BPA-exposed cells identified 551 differentially expressed (DE) transcripts (RNA-seq FPKM values ≥ 2-fold change [FC], *p*-value ≤ 0.01) (Figure 2c). All 551 DE transcripts (244 upregulated and 307 downregulated transcripts) in the BPA-treated Y79 cells are listed in Appendix A. The result was corroborated by the heatmap representation based on the hierarchical clustering of expression ratios for the DE transcripts (Figure 2d).

### 3.3. Gene Ontology (GO) Analysis Revealed That BPA Treatment Changed Gene Expression Sets Related with RNA Processing

To identify the transcriptomic pathways affected by BPA exposure, the subset of DE transcripts that were significantly affected by the BPA treatment were subjected to GO annotation using the DAVID bioinformatics resource (https://david.ncifcrf.gov (accessed on 5 February 2021)). For each category, these results were defined to be statistically significant at *p* < 0.001. The analysis was performed to identify GO pathways at biological processes and molecular function categories (Appendix A). The upregulated and downregulated genes were independently subjected to GO analysis to distinguish them according to their functional roles (based on their expression patterns) and not merely according to their gene names. The biological processes, which were significantly enriched, were mainly involved in the negative regulation of biological processes, negative regulation of cellular processes, regulation of catalytic activity, and regulation of hydrolase activity, and the post-transcriptional regulation of gene expression categories were significantly clustered (*p* < 0.001) (Figure 3a). The most DEG (differentially expressed genes)-affected molecular functions were associated with nucleic acid binding, RNA binding, and tRNA binding (Figure 3b).

### 3.4. Abnormal Alternative Splicing Events Were Markedly Increased by BPA Treatment in Y79 Retinoblastoma Cells

Alternative splicing plays an important role in a tightly regulated process that dramatically expands the diversity of the transcriptome and proteome encoded by the genome [17]. Abnormal alternative splicing events are emerging as a hallmark of various diseases such as cancer [18]. In this study, we investigated whether BPA treatment in Y79 cells affected alternative splicing (AS) events. Because exposure to BPA changed the expression of gene sets of RNA maturation and splicing, we focused on the BPA-dependent alternative splicing (AS) events in RNA-seq data.

A comparison of all RNA-seq data from the control versus BPA-treated group identified differential AS events for all five major types, including skipped exon (SE), alternative 5′-splice site exons (5′-SS), alternative 3′-splice site exons (3′-SS), mutually exclusive exons (ME), and retained intron (RI) (FDR < 0.01, FPKM > 1; Figure 4a). We used the MISO software to detect the differential exon usage in the BPA-exposed group. The identified splice site variants were categorized as 86,017 SE events, 7378 5′-SS events, 11,214 3′-SS events, 11,792 ME events, and 7152 RI events (Figure 4a, left panel). These events showed the distribution of protein-coding genes (Figure 4a, right panel). Over 80% of protein-coding genes in the BPA-exposed group were involved in SE events. Interestingly, 3971 of retained intron (RI) events (55.52% of whole RI events) were included in protein-coding genes (Figure 4a). RI is exemplified by a component of the spliceosome Mago Homolog B, Exon Junction Complex Subunit (*MAGOHB*) (Figure 4b) and RNA binding protein Heterogeneous Nuclear Ribonucleoprotein D (*HNRNPD*) (Figure 4c). The *MAGOHB* transcript in the control group did not include any intron sequences; however, the intron was not appropriately removed in the BPA-treated group. Besides, intron of the *HNRNPD* transcript was not removed in the BPA-treated group, suggesting a novel effect of BPA on transcript maturation. We also found several alternative transcript isoforms that were regulated by BPA treatment. ATP binding cassette subfamily F member 3 (*ABCF3*, Appendix A), Histone Deacetylase 1 (*HDAC1*, Appendix A), RELA proto-oncogene (*RELA*, Appendix A), Heterogeneous Nuclear Ribonucleoprotein M (*HNRNPM*, Appendix A), Fused in sarcoma (*FUS*, Appendix A), and RNA Binding Motif Protein 3 (*RBM3*, Appendix A) genes have differentially expressed transcripts suggesting dynamic transcriptional regulation by the BPA treatment.

### 3.5. Adverse Effects of BPA Revealed by Proteomic Analysis in Y79 Retinoblastoma Cells

To further investigate the molecular profiling of BPA-treated Y79 cells, we identified the protein expression patterns using 2D-electrophoresis and MALDI-TOF-MS analysis. The tested spots were mapped onto analytic gels that were stained with silver. Based on these analyses, the differentially expressed protein spots were excised and subjected to in-gel digestion followed by peptide mass fingerprinting for protein identification (Figure 5a). The NCBI human database (Homo sapiens, 227776 sequences) was searched using MASCOT software, and 38 differentially expressed proteins were evaluated by MALDI-TOF-MS analysis. For each identified protein, a probability-based score greater than 66 was considered significant (*p* < 0.05). A comparison of the abundant protein expression pattern between the control and BPA-treated groups was performed using the PDQuest software program (Figure 5a). Nine proteins were identified that were significantly upregulated or downregulated by BPA exposure (Figure 5b, Appendix A). The identified proteins, including PDIA5, HYOU1, MTHFD2, LDHA, TBP, PARK7, GAPDH, and VCP, were upregulated by BPA exposure, and only one protein, HIST2H4B, was downregulated by BPA exposure in Y79 cells (Figure 5c). Furthermore, we investigated the protein interaction relationship within identified protein set using STRING 11.0 (http://string-db.org (accessed on 5 February 2021)). The obtained protein interaction network was composed of nine nodes connected by 12 edges with a PPI enrichment *p*-value < 7.95 × 10^−5^. We analyzed this protein network by using the KEGG pathway and obtained three network modules associated with KEGG pathways such as glycolysis/gluconeogenesis, HIF-1 signal pathway, and protein processing in the endoplasmic reticulum. The glycolysis/gluconeogenesis (FDR < 0.01), HIF-1 signal pathway (FDR < 0.01) were significantly associated with the GAPDH, LDHA protein network. In addition, interaction between HYOU1 and VCP was associated with protein process in ER (FDR < 0.01), suggesting the effect of BPA on regulating cell metabolism (Figure 5d) Among them, we further evaluated the expression levels of HYOU1 through Western blot analysis. As shown in Figure 5e, the expression level of HYOU1 was upregulated in the BPA-exposed group compared to control in Y79 cells.

## 4. Discussion

Exposure to BPA has been associated with various harmful effects on the human body [9]. However, to our knowledge, this was the first study to report the association of BPA with the retinal cellular environment. This study aimed to determine whether low-dose BPA exposure is associated with changes in comprehensive gene expression and protein levels using retinoblastoma Y79 cells. Although the data from the human retinoblastoma cell line Y79 are thought to be limited in interpreting the effects of BPA on the human eye, it will be a new result that can confirm changes in the molecular level of the eye exposed to contaminants such as BPA. In general, toxicants, such as bisphenols, lead to molecular-level changes in the cellular environment, we need to identify the adverse effect of BPA early before apoptotic cell death and physiological changes. However, we found that the highest concentration of BPA caused molecular damage to Y79 cells (Figure 1, Appendix A). Thus, we investigated the alterations of transcripts and protein levels associated with BPA exposure, revealing a minimal effect on the molecular behavior of Y79 cells.

For several decades, it has been reported that BPA has a detrimental effect in vitro and in vivo [9,19,20,21]. Thus, some countries have banned the usage of BPA in several personal products. Nevertheless, over 6 billion pounds of BPA are produced annually through extensive industrial applications of polycarbonate and epoxy resins, and 100 t of BPA is emitted into the atmosphere every year [22]. People are concerned about how this kind of air pollution affects human organs, including the eyes. Thus, several studies have reported the adverse effects of air pollution on human eyes. For example, Chua et al. (2020) identified a relationship between ambient air pollution and eye dysfunction using UK Biobank data. They reported that all of the air pollutants, such as PM2.5, PM10, and NOx, influence the apparent features of retinal structure. Additionally, scientists from AIIMS, New Delhi, India demonstrated that 10–15% of people in Delhi had problems with chronic irritation and dry eye syndrome due to long-term exposure to the high levels of air pollution [23,24]. Despite the increasing evidence that air pollution has a detrimental effect on the human eye, the relevance of ocular exposure to BPA, in particular, remains unclear. However, there is limited information on the transcriptome changes associated with low-dose BPA in developmental retinal cells. To explore the diverse biological responses to BPA, we performed comparative analyses of comprehensive gene expression by RNA-seq performed in Y79 cells treated with 40 μM BPA for 48 h. Our findings from RNA-seq analysis of human retinoblastoma cells exposed to BPA suggest that BPA disrupts post-transcriptional regulation of gene expression.

Recently, it has been reported that differentially expressed proteins translated from alternative splicing variants are implicated in human diseases [25,26,27]. In the presented results, we identified differential alternative splicing events for all five types, including SE, 5′SS, 3′SS, ME, and RI (Figure 4a) [17,28]. Among these events, retained intron (or intron retention (IR)) is a pivotal player in the fine-tuning gene expression at the post-transcription regulation during normal development, including translational inhibition in response to hypoxic stress [29], gene regulation during hematopoiesis [30,31], and neurogenesis [32]. Accumulated evidence indicated that RI has been widely regarded as an aberrant splicing event that is associated with various diseases [33]. For instance, abnormal expressed RI is one of the regulators of transcriptome diversity in cancer [34] and is a causative factor in the inactivation of different tumor-suppressor genes [35]. Despite causing abnormal gene expression by RI, there is no evidence for the effect of BPA on expression of RI, at least in this cell lines. Thus, we focused on RI induced by BPA treatment. In present study, we found that the identified genes, including *MAGOHB*, *HNRNPD*, *ABCF3*, *HDAC1*, *RELA*, *HNRNPM*, *FUS*, and *RBM3*, were altered by exposure to BPA in Y79 cells (Figure 4). These genes play a role in various cellular processes, such as transcription regulation, RNA splicing, RNA transport, and nonsense-mediated decay (NMD). *MAGOHB* encodes proteins associated with core members of the exon junction complex. This gene is required for pre-mRNA splicing as a component of the spliceosome that mediates various downstream processes for RNA metabolism (NMD) [36,37]. The differential gene expression pattern of *MAGOHB* has been implicated in breast cancer development [38]. Furthermore, *HNRNPD* and *HNRNPM* encode the subfamily of ubiquitously expressed heterogeneous nuclear ribonucleoproteins (hnRNPs). *HNRNPD* is a critical factor for RNA-destabilizing factor in the ARE-mediated decay pathway [39]. Dysfunction of hnRNPD has been implicated in tumorigenesis [40]. *HNRNPM* plays roles in exon skipping/inclusion [41] and the regulation of the innate immune response to infection [42]. Furthermore, *ABCF3* is a member of the ATP-binding cassette (ABC) transporter superfamily. This gene mediates flavivirus resistance in the presence of a binding partner, OAS1B [43]. HDAC1 is well known as a component of the histone deacetylase complex. HDAC1 acts as a repressor in cell proliferation and differentiation [44]. RELA, also known as p65, is a transcription factor that presents most cell types and participates in various biological processes, such as inflammation, immunity, differentiation, cell growth, tumorigenesis, and apoptosis [45]. Additionally, FUS is a multifunctional DNA/RNA binding protein, and this gene is associated with familial amyotrophic lateral sclerosis/frontotemporal dementia [46,47]. FUS has also been verified as a splicing regulator based on its presence in spliceosomal complexes [48]. Finally, RBM3 is a cold-inducible RNA binding protein that is involved in mRNA biogenesis, wherein it manifests an anti-apoptotic effect. The RBM3 gene is a proto-oncogene that is involved in tumor progression and metastasis [49]. These genes altered by BPA are involved in the post-mitotic pathway. The presented results suggest that BPA does not exert a powerful cytotoxic effect, but it causes changes in the normal transcriptome profile, which is a potential cause of disease. Additionally, multi-dimensional epigenomic analysis could provide the insights to understand the adverse outcome pathway by the chemical exposure [50]. A previous study demonstrated the effect of the environmental chemical, methylparaben, on the transcriptome profiling of human non-small cell lung carcinoma cells [51]. Methylparaben was shown to alter gene expression, including that of *FUS*. In the present study, we also identified changes in several gene expression pathways, including that of *FUS*. However, we could not find any further published evidence to corroborate our observations regarding the effects of environmental pollutes. We demonstrated that alternative splicing events and altered protein expression patterns are critical phenomena affected by low-dose BPA exposure.

We further demonstrated the alteration of protein expression patterns using proteomic profiles on Y79 cells (Figure 5). The identified nine proteins, which are significantly upregulated or downregulated by BPA exposure, were analyzed by their protein–protein interaction and additional analysis using the KEGG pathway (Figure 5). Especially, it has been shown that HYOU1 is one of the upregulated genes by BPA and is related to cellular metabolisms. HYOU1 is also known as oxygen-regulated protein 150 (ORP150). This protein localizes to the endoplasmic reticulum and mitochondria, and it protects cells from cell stress responses, including oxidative stress and unfolded protein responses [52,53]. Yang et al. has recently demonstrated that mitochondria dysfunction and endoplasmic reticulum stress bornyl cis-4-hydroxycinnamate exposure occurs upregulated HYOU1 expression in melanoma cells leading to induction of apoptosis [54]. Furthermore, it has been demonstrated that HYOU1 is related to various pathologic situations, such as ischemic brain [55], malignant tumors [56]. Furthermore, it has been shown that HYOU1 is upregulated in ischemic condition of mouse retina [57].

As is known, the eye is one of the human organs under chronic oxidative stress due to the lifelong exposure to light, consumption of high oxygen, and partial pressure of high oxygen from the underlying basal capillaries. This oxidative stress condition is thought to contribute to retinal disease. Indeed, it was suggested that oxidative stress influences the pathophysiological function in the elderly blinding disease age-related macular degeneration (AMD) [58]. It is well known that BPA is a causative factor to the impairment of redox homeostasis through increasing mitochondrial dysfunction and induction of apoptotic cell death [59]. In addition, a recent study has reported that endocrine disruptors such as BPA may contribute to the increase in eye diseases [60]. With these results, HYOU1 induced by BPA treatment might be a potential therapeutic target for eye diseases. As shown in this study, long-term exposure to harmful environmental pollutants and resultant altered molecular behavior in the human body may potentiate epigenetic changes in transcriptome and proteome profiles. Various alternative splicing events and altered protein expression patterns will be potential targets for novel biomarkers used for the early detection of various diseases associated with environmental pollution.

## 5. Conclusions

In this study, we investigated the cellular cytotoxicity and expression pattern by BPA exposure on human retinoblastoma cells. BPA did not show adverse effect, such as cell viability or cell death. Furthermore, we conducted a comprehensive transcriptomic and proteomic profiling by low doses of BPA at long-term exposure. Transcriptome analysis using RNA-seq identified the alternation of post-transcriptional regulation-associated gene sets and cell cycle regulation-associated gene sets. Additionally, we found the differential alternative splicing events. Particularly, retained intron (RI) event induced by BPA revealed changes in *MAGOHB*, *HNRNPD*, *ABCF3*, *HDAC1*, *RELA*, *HNRNPM*, *FUS*, and *RBM3*. In proteome analysis using MALDI-TOF-MS revealed a total of nine differentially expressed proteins (PDIA5, HYOU1, MTHFD2, LDHA, TBP, PARK7, GAPDH, VCP, and HIST2H4B). This study provides a novel marker for the detection of various diseases associated with environmental pollutants such as BPA.

## Figures and Tables

**Figure 1 genes-12-00264-f001:**
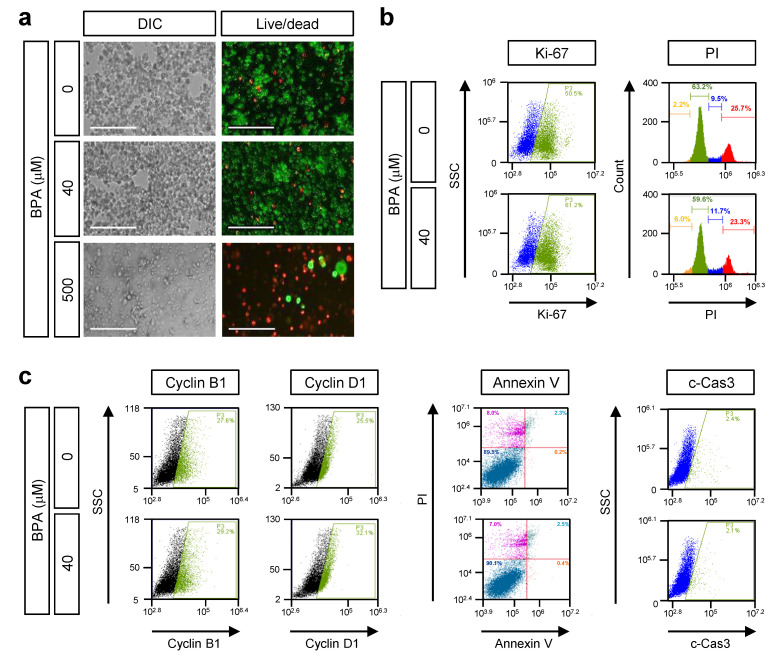
Cytotoxic effect of bisphenol A (BPA) on Y79 retinoblastoma cells. Cells were treated with the indicated concentration of BPA for 48 h. (**a**) Morphological changes in Y79 cells were observed by phase-contrast microscopy after BPA treatment. Cell viability was analyzed using the live/dead assay. Fluorescence microscopy images of live or dead cells after 30 min staining with calcein-AM (0.3 μM, Live cells) and EthD-1 (3 μM, Dead cells). Scale bars = 200 μm. (**b**) Cellular behavior by BPA treatment measured using FACS analysis. Cells were stained antibodies against each markers; Ki-67 for cell proliferation, propidium iodide (PI) for cell cycle. (**c**) Cyclin B1 and cyclin D1 for detection of cell cycle markers, Annexin-V/PI staining for apoptosis, cleaved Caspase-3 for detection of the active apoptosis. Values are presented as mean ± S.E.M., n = 6.

**Figure 2 genes-12-00264-f002:**
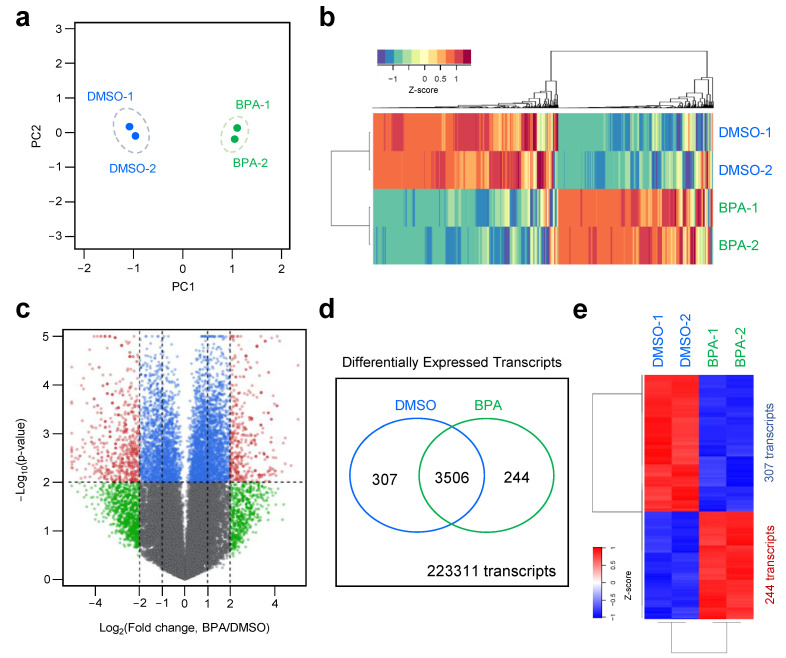
Dynamic transcriptome profiles by the BPA treatment in Y79 retinoblastoma cells. Comparative analysis of transcriptome generated by directional RNA-seq from BPA treated Y79 cells. (**a**) Principal component analysis (PCA) of directional RNA-seq data. The values reveal the amount of variation attributed to each principal component. Small circles indicate individual samples, and larger ones show each group. (**b**) Transcriptional pattern analysis in control and BPA-exposed groups by employing heat map and hierarchical clustering. (**c**) Volcano plot shows differentially expressed transcripts in BPA treated group compared to the DMSO-treated group. Small red circled transcripts were verified as having the absolute value of log2 (fold changes) and −log10 (*p*-value). (**d**) Differentially expressed transcripts were identified by the DEseq in BPA treated group from total annotated transcripts. (**e**) Heatmap depicting fold changes for all transcripts indicating statistically significant differences between the BPA-treated group and the control group.

**Figure 3 genes-12-00264-f003:**
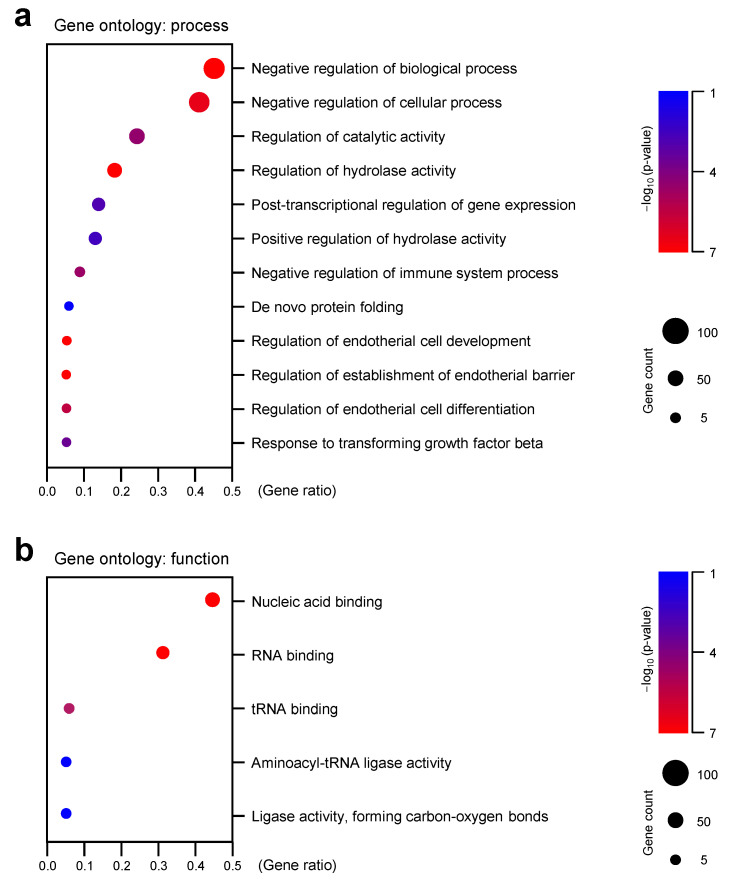
Analysis of gene ontology in cells treated with BPA. The classification of gene ontology (GO) functional enrichment analyses with the differentially expressed genes (DEGs) from comparisons of control and BPA-treated Y79 cells. The functional enriched classes of downregulated DEGs annotated by biological process (**a**) and molecular function (**b**).

**Figure 4 genes-12-00264-f004:**
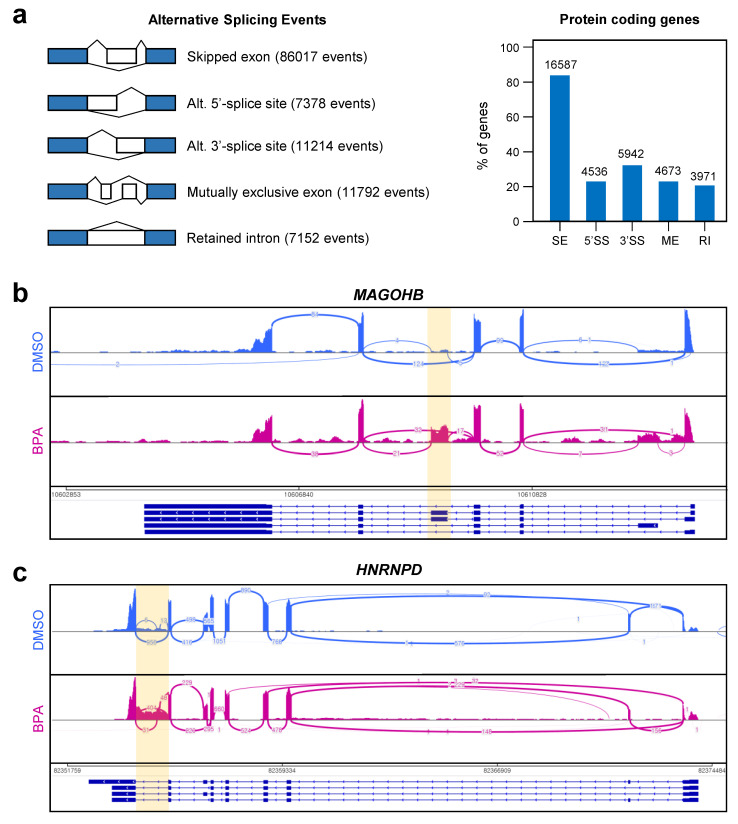
Transcriptome dynamics identifies the alternative splicing events in BPA-treated Y79 cells. (**a**) Alternative splicing events were analyzed by the MATS and identifies alternative splicing events. Types of alternative splicing events detected and counted their frequencies in group-wise comparisons. Protein coding genes obtained from alternative splicing events indicated as the percentage of genes. Differential splicing events in the BPA-treated group shown the clear retained intron (RI) in Mago Homolog B, Exon Junction Complex Subunit (*MAGOHB*) (**b**) and Heterogeneous Nuclear Ribonucleoprotein D (*HNRNPD*) (**c**) genes. The yellow shadow indicates the retained intron region in BPA-treated Y79 cells.

**Figure 5 genes-12-00264-f005:**
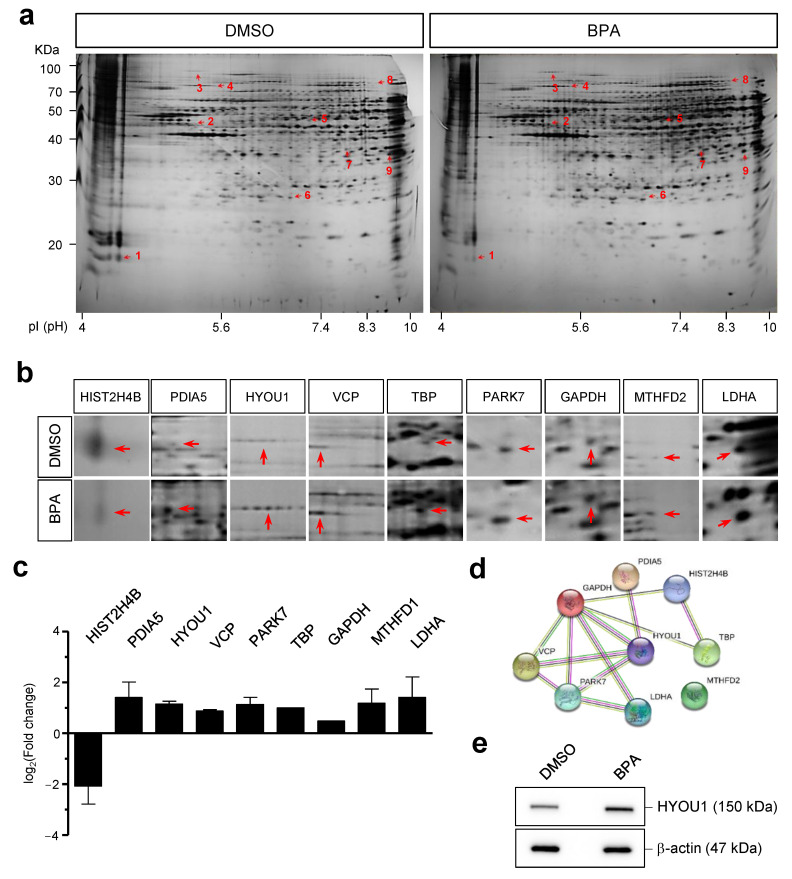
Comparative proteome analysis by low-dose of BPA in Y79 cells. (**A**) Representative silver staining images of 2D electrophoresis patterns from BPA-treated and control groups in Y79 cells. Each spots indicate differentially expressed proteins (red arrow), up- and downregulated proteins, respectively. HIST2H4B (1), PDIA5 (2), HYOU1 (3), VCP (4), PARK7 (5), TBP (6), GAPDH (7), MTHFD2 (8), LDHA (9). (**B**) The nine proteins identified that significant up- and downregulated by exposure to BPA. (**C**) Selected proteins are represented as fold changes (log_2_). (**D**) The protein network was identified using the STRING database. Network nodes represent proteins shown by gene names and colored lines represent protein–protein interactions. (**E**) Western blot analysis of HYOU1. β-actin was used as a protein loading control.

## Data Availability

The data presented in this study are available in the article or Appendix A.

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
