# Peer review of "Bisphenol A Exposure Changes the Transcriptomic and Proteomic Dynamics of Human Retinoblastoma Y79 Cells"

_genes, 2021, doi:10.3390/genes12020264_

Round 1

Reviewer 1 Report

It is really great manuscript to talk about the possible toxicity of BPA which is generally used many manufactural products. Especially the alternative slicing events which might be missed by conventional analyzing method. Plus, proteome profiling can reconfirm the potential toxicity found by RNA-seq analysis.

I think author needs to provide or explain the detail information of the control in Figure S1-A. DMSO itself is toxic and high volume of that can kill the cells. How much volume of DMSO alone was used for control in this MTT assay.

I could not see the histogram peak of 'apoptosis' and 'S-phase' inFigure S1-C, cell cycle assay using PI. The resolution of the FACS data is generally low, too.    I think this is because of transition of the data as 'Microsoft Word' file in which the format is not matched. I am sure that author submitted the good quality of pictures but there was something happen while the file was changed as different format. Although the supplementary figures are low resolution, the manuscript (pdf file) figures are pretty good resolution. that's why I think the format issue in this problem.

In Figure S2-A and S2-B, the positive cells are so few. it seems like the cells were either not permeabilized for staining or the cells were not growing at all. In case of later, it is better to drop this data.

In figure S3-A, the apoptosis assay using Annexin-V antibody showed something weird pattern. I am sure that the data needs compensation for proper pattern of apoptosis analysis because author used FITC conjugated Annexin-V antibody with PI which is required to compensate each other.

Reviewer 2 Report

Kim et al. analyzed a quite interesting topic and provide new information and perspective regarding BPA exposure and human retinoblastoma. They followed a, in my opinion, complete experimental workflow, which contained analysis in cell properties, RNA splicing and protein expression levels. Therefore, I believe that this article merits publication in Genes, providing the following issues are addressed:

  • In the experimental workflow only one cell line has been used and therefore I believe that this should be mentioned in the manuscript, as one of the limitations of the study.
  • The information of figures 2B, 2C and 2E overlap with each other and I suggest that only one of the figures to be kept in the manuscript.
  • Figure 5A is not large enough and the green spots are easily seen. Moreover, it the difference between the two panels in Figure 5A is not obvious. The size of Figure 5B is also not large enough and therefore the differences between the two panels in each row are not obvious.
  • Why did the authors choose to further investigate with Western Blot only HYOU1?
  • The final sentences if section 3.5 have been merged with the legend of Figure 5. Please correct this.
  • The interaction network of HYOU1 cannot be seen in the Figure 5E. Please correct this figure.
  • In the section of discussion, the following sentence is mentioned “Among these events, we focused on RI induced by 430 BPA treatment”. Why did the authors focus only on these alternative splicing events?
  • The section of discussion is not well-organized. The authors mention their key findings but do not further analyze them. Additionally, they should underline their value in the context of BPA and human retinoblastoma.
